# Dynamic Changes in Flavor and Microbiota in Traditionally Fermented Bamboo Shoots (*Chimonobambusa szechuanensis* (Rendle) Keng f.)

**DOI:** 10.3390/foods12163035

**Published:** 2023-08-12

**Authors:** Zhijian Long, Shilin Zhao, Xiaofeng Xu, Wanning Du, Qiyang Chen, Shanglian Hu

**Affiliations:** 1School of Life Science and Engineering, Southwest University of Science and Technology, Mianyang 621010, China; long20053182@swust.edu.cn (Z.L.); z740506287@126.com (S.Z.); 18788981570m@sina.cn (X.X.); dwncie@sina.com (W.D.); chenqiyang@swust.edu.cn (Q.C.); 2Engineering Research Center for Biomass Resource Utilization and Modification of Sichuan Province, Mianyang 621010, China

**Keywords:** sour bamboo shoot, microbial compositions, traditional fermentation, volatile flavor compounds, correlation analysis

## Abstract

Dissecting flavor formation and microbial succession during traditional fermentation help to promote standardized and large-scale production in the sour shoot industry. The principal objective of the present research is to elucidate the interplay between the physicochemical attributes, flavor, and microbial compositions of sour bamboo shoots in the process of fermentation. The findings obtained from the principal component analysis (PCA) indicated notable fluctuations in both the physicochemical parameters and flavor components throughout the 28 day fermentation process. At least 13 volatile compounds (OAV > 1) have been detected as characteristic aroma compounds in sour bamboo shoots. Among these, 2,4-dimethyl Benzaldehyde exhibits the highest OAV (129.73~668.84) and is likely the primary contributor to the sour odor of the bamboo shoots. The analysis of the microbial community in sour bamboo shoots revealed that the most abundant phyla were Firmicutes and Proteobacteria, while the most prevalent genera were *Enterococcus*, *Lactococcus*, and *Serratia*. The results of the correlation analysis revealed that Firmicutes exhibited a positive correlation with various chemical compounds, including 3,6-nonylidene-1-ol, 2,4-dimethyl benzaldehyde, silanediol, dimethyl-, nonanal, and 2,2,4-trimethyl-1,3-pentylenediol diisobutyrate. Similarly, *Lactococcus* was found to be positively correlated with several chemical compounds, such as dimethyl-silanediol, 1-heptanol, 3,6-nonylidene-1-ol, nonanal, 2,2,4-trimethyl-1,3-pentanediol diisobutyrate, dibutyl phthalate, and TA. This study provides a theoretical basis for the standardization of traditional natural fermented sour bamboo production technology, which will help to further improve the flavor and quality of sour bamboo.

## 1. Introduction

Bamboo shoots, the young culms of bamboo plants, are rich in protein, fiber, vitamins, minerals, and phytosterols [1]. Because bamboo shoots are highly moist and lignify quickly, they are hard to store for a long time. The fermentation of bamboo shoots into sour bamboo shoots makes them palatable in flavor, aroma, texture, and appearance. Additionally, fermentation promotes the diversity of bamboo shoot products and extends their shelf life [2].

In China and southeast Asia, fermented bamboo shoots have become increasingly popular due to their unique taste and texture [3]. The critical step in processing sour bamboo shoots is to ferment them by soaking them in cold boiling water or mountain spring water for a certain period (15–30 days) [1]. Under the influence of microorganisms, bamboo shoots gradually turn sour, and the dominant flora and flavor-related flora are diverse among different fermented bamboo shoot varieties. Among sour bamboo shoots in Guangdong, *Pichia*, *Candida*, and *Debaryomyces* dominate, while *Pichia* and *Zygosaccharomyces* dominate in Yunnan [3]. The main genera found in Yunnan sour bamboo shoots (*Dendrocalamus latiflorus* Munro) are *Lactococcus* and *Lactobacillus* [4]. The fermentation of Guanxi sour bamboo shoots (*Dendrocalamus latiflorus* Munro) began with *Weissella*, then *Lactobacillus* increased continuously, finally becoming the dominant bacterial species [5]. When fresh Ma bamboo shoots are fermented for 30 days, *Lactobacillus*, *Clostridium_sensu_stricto_1*, *Enterobacter*, and *Leuconostoc* exhibit a significant association with sour bamboo shoot flavor. As a characteristic flavor compound of sour bamboo shoots, p-cresol is highly correlated with *Clostridium* [6]. Li, et al. [7] have found that *Lactobacillus*, *Lactococcus*, *Weissella*, and *Cyanobacteria* dominate the bacterial community in Ma bamboo shoots after 35 days of fermentation. According to the research findings of Xia, et al. [8], several types of halophilic bacteria were seen to be the predominant strains during the initial three days of fermentation whereas *Lactococcus lactis* and *Weissella* sp. were seen to take over as the dominant strains after seven days of fermentation. These studies enhance our understanding of the succession of microbial communities throughout the fermentation process. However, there is still a lack of clarity when it comes to identifying the specific compounds that contribute to the notably unpleasant odor of sour bamboo shoots, as well as the microorganisms that are responsible for their fermentation. Additionally, the diverse microbial community present in sour bamboo shoots yields a complex and variable flavor composition [3]. To achieve a comprehensive understanding of fermentation processes and ensure the consistent quality of fermented products, it is essential to investigate the succession and function of the microbes responsible for the formation of characteristic flavor compounds.

Consequently, the current investigation involved acquiring freshly harvested bamboo shoots (*Chimonobambusa szechuanensis* (Rendle) Keng f.) sourced from the southwestern region of China. Following this, the shoots were subjected to a 28 day fermentation process to prepare sour bamboo shoots. During the sour bamboo shoot fermentation process, flavor compounds (organic acids and volatiles), physicochemical properties (pH, titratable acidity, nitrite, and reducing sugar content), and texture properties were analyzed at 1, 7, 14, 21, and 28 days. In addition, high-throughput sequencing of 16S rRNA was used to study microbial communities and observe their dynamic changes. As part of this study, we aim to gain a better understanding of the fermentation mechanisms underlying the production of sour bamboo shoots by examining the associations between physicochemical properties, dominant microbes, and major flavor compounds, and by providing a theoretical basis for the standardization of traditional production methods.

## 2. Materials and Methods

### 2.1. Experimental Materials

Fresh bamboo shoots (*Chimonobambusa szechuanensis* (Rendle) Keng f.) were collected from Leshan (Sichuan, China) in October 2022, and delivered to the laboratory premises within 24 h. The bamboo shoots were then carefully cleaned by removing their shells and washing them in clean water. Ultimately, the central edible component of the bamboo shoots was retained for further processing.

### 2.2. Fermentation Process

Before the bamboo shoots were used, they were washed and drained. The bamboo shoots (shelled) were sliced into julienne strips that were 13 cm long, 0.5 cm wide, and 0.5 cm thick. After placing the strips in a plastic bottle filled with sterile water (450 mL), the bottle was sealed and left to ferment at room temperature (22 ± 2 °C) for 28 days. Preparation and monitoring of 15 bottles were conducted. On days 1, 7, 14, 21, and 28, samples were taken randomly from three unopened bottles each time. In addition to the immediate analysis of the bamboo shoots’ texture, fermentation liquids were stored at −80 °C for further analysis.

### 2.3. Texture Analysis

A texture analyzer (TA-XT. Plus, Stable Micro Systems Ltd., Godalming, UK) was used to measure changes in hardness, friability, and chewiness of sour bamboo shoots during fermentation. A P36R probe was used to test texture properties on consistently sized sour bamboo shoots (30 mm × 5 mm × 5 mm). During the experiment, the parameters were set on a pre-test speed of 2 mm/s, an initial test rate of 1 mm/s, an initial post-test rate of 1 mm/s, a compression rate of 70%, a pause of 5 s, and a trigger force of 5 g.

### 2.4. Determination of pH and Total Acid (TA)

A pH meter (PHS-25, Greifensee, Switzerland) was used to measure the pH value. TA measurements were conducted following GB/T 12456-2021 [9]. To determine TA, 10 mL of fermentation liquid was titrated with 0.1 N NaOH and 0.5 mL of phenolphthalein was added as an acid–base indicator.

### 2.5. Determination of Reducing Sugar Content

The reducing sugar content of the fermentation liquids was measured according to GB 5009.7-2021 [10].

### 2.6. Determination of Nitrite Content

Nitrite content in fermentation liquids was determined using ultraviolet spectrophotometry as described by GB 5009.33-2016 [11]. The nitrite concentration in the fermentation liquid was determined by adding sulfonamide and N-(1-Naphthyl) ethylenediamine dihydrochloride (NED) solutions. At room temperature, the mixtures were incubated for 15 min. Samples were then measured at 538 nm for absorbance. To calculate nitrite, a NaNO_2_ standard curve was used (0.02–0.25 mg/L), and the results are expressed in mg/kg. A triplicate of each determination was performed.

### 2.7. Determination of Volatile Compounds

Headspace solid-phase microextraction technique and gas chromatography-mass spectrometry (HS-SPME-GC-MS/MS, Varian450GC, Varian Analytical Instruments, Harbour, CA, USA) were employed to identify the volatile compounds present in the fermentation liquids at various stages of fermentation. As per the methodology, adjusted slightly, of Chen, et al. [4], the sealed headspace vial housed 8 mL of sour bamboo shoot fermentation broth and 3 g of NaCl. The vial was subsequently immersed in a water bath set at a temperature of 60 °C for 10 min. The volatile components were then effectively collected through the use of divinylbenzene/carboxy/polydimethylsiloxane (/CAR/PDMS/DVB, 65 μm) fibers (Supelco Inc., Bellefonte, PA, USA), with the absorption process lasting 30 min. Lastly, the fibers containing the volatiles were subjected to desorption, which was performed on a heated GC jet at a temperature of 250 °C for 5 min. To quantify the volatile compounds, 100 μL n-hexane containing 38.12 μg cyclohexanone was added as an internal standard. Volatile compounds were analyzed using a GC-MS with a flexible capillary column (HP-5MS, 30 m × 0.25 mm × 0.25 μm). Helium gas was used as a carrier gas, with a constant flow rate of 1.0 mL/min and no split flow. The temperature profile for the GC oven was as follows: starting at 40 °C, it was increased at a rate of 12 °C/min to reach 100 °C, followed by a ramp of 5 °C/min to 120 °C. Thereafter, the temperature was increased to 180 °C at the rate of 8 °C/min and held for 5 min. The final temperatures were then reached, in increments of 12 °C/min to 210 °C and 8 °C/min to 250 °C. The mass spectrometer was operated at a source temperature of 230 °C, an ionization potential of 70 eV, and a mass scan range (*m*/*z*) of 40–400 amu. These parameters were consistently maintained throughout the experiment. We calculated the odor activity value (OAV) of each compound by dividing the concentration of volatile compounds by the threshold to evaluate the contribution of the compound to the formation of a sour bamboo aroma. Compounds with OAV ≥ 1 are believed to be primary contributors to the flavor of sour bamboo shoots [12].

### 2.8. Microbial Analysis

Genomic DNA from fermentation liquid was isolated by cetyltrimethylammonium bromide (CTAB). Nanodrop and agarose gel electrophoresis were used to determine the purity and concentration of DNA. The obtained DNA was subsequently amplified with 338F (5′-ACTCCTACGGGAGGCAGCA-3′) and 806R (5′-GGACTACHVGGGTWTCTAAT-3′) universal primer sets, targeting the V3–V4 regions of the 16S rRNA gene. Samples exhibiting a brightness reading between 400 and 550 bp were selected for sequencing. A 250 bp double-end sequencing read was obtained using Illumina platform following quality control [13]. By filtering the raw reads obtained from sequencing with Trimmatic v0.33, the results of the information analysis become more accurate and reliable. Subsequently, cutadapt 1.9.1 software was employed to identify and eliminate primer sequences to obtain clean reads devoid of primer sequences. The dada2 method in QIIME2 2020.6 was used for denoising [14,15]; here, the two-ended sequences must be concatenated while the chimeric sequences should be discarded to obtain the final valid data or non-chimeric reads. Sequences showing ≥97% similarity were grouped into operational taxonomic units (OTUs). NMDS and PCoA can be used to examine differences in community structure across samples or groups. LDA effect size (LEfSe) was used to quantify the differences in community structure between grouped samples.

### 2.9. Statistical Analysis

A statistical analysis was performed using SPSS Statistics 25.0 (IBM, Chicago, IL, USA). One-way analysis of variance (ANOVA) with Tukey’s honest significant difference (HSD) post-hoc test was utilized to determine significant differences among the groups. In addition, principal component analysis (PCA) and Pearson correlation were used to investigate the interaction between physiochemical characteristics, flavor, and microorganisms. All values are represented as mean ± standard deviation (SD) with *p* < 0.05 considered statistically significant.

## 3. Results and Discussion

### 3.1. Physiochemical Property Analysis

As shown in Table 1, sour bamboo shoots undergo fermentation by gradually lowering and stabilizing pH levels. After 14 days of fermentation, there was no significant variation in the pH value (4.92–4.81) of sour bamboo shoots, which is consistent with the reported trend of fermented bamboo shoots (*Dendrocalamus latiflorus* Munro) [16]. The acidic environment during fermentation may be attributed to the acid or other acidic compounds produced by *Bacillus* and *Lactobacillus*, as suggested by Jeyaram, et al. [17]. Such acidic conditions throughout the fermentation stage are advantageous in enhancing the shelf life and quality of sour bamboo shoots. The content of TA increased steadily from 2.50 ± 0.43 g/L to 8.75 ± 0.29 g/L (Table 1), consistent with that of Ma sour bamboo shoots [6]. The recorded nitrite measurements showed a marked increase from an initial level of 0.12 ± 0.01 mg/kg on day 1 to a peak level of 0.75 ± 0.08 mg/kg. Subsequently, there was a significant decrease, with measurements registering at 0.19 ± 0.03 mg/kg on day 28. A nitrate-reducing bacteria reduces nitrate during the early stages of fermentation, resulting in an increase of nitrite content. Nitrate-reducing bacteria are usually acid-intolerant microorganisms such as *Escherichia coli* and *Pseudomonas* spp. carried by bamboo shoots. The growth of these bacteria is gradually inhibited as fermentation proceeds, and the amount of nitrites decreases. Moreover, under acidic conditions, NO_2_^−^ can combine with H^+^ to form HNO_2_, and then self-disproportionation occurs to produce nitrogen dioxide and nitric oxide, thus decreasing nitrite. [18]. In compliance with the national health standard level for nitrite (20 mg/kg) in China, the sour bamboo shoots featured in this study are safe for human consumption [19]. In addition, it was recorded that the maximum level of reducing sugar content occurring on day 14 was 1.18 ± 0.02 g/L, followed by a gradual reduction in the reducing sugar content, primarily owing to microbial consumption [6].

The evaluation of quality in sour bamboo shoots is greatly influenced by their texture, particularly regarding their hardness, fracturability, and chewiness [20]. During the first 7 days of fermentation, there was a significant decrease in hardness, fracturability, and chewiness of fermentation, after which the decline was relatively gradual (Figure 1). A similar softening phenomenon also occurred in Sichuan pickles [21] and pickled chayote [22]. An analysis of physiochemical indexes using Pearson’s correlation (Appendix A) revealed a positive correlation between pH and hardness, fracturability, and chewiness. The pH level was lowered, resulting in a decrease in the permeability of cell membranes, which will negatively affect textural characteristics [4]. Additionally, microorganisms involved in the fermentation process have the ability to generate pectinase and cellulose, both of which can break down cell walls and destroy the cellular structure, ultimately impacting the hardness, fracturability, and chewiness of the sample [23].

### 3.2. Volatile Compounds of Sour Bamboo Shoots

During the entire fermentation process of sour bamboo shoots, a total of 43 volatile compounds were detected (Appendix A), among which there were at most 17 types of alcohol volatile compounds, followed by ketones and esters (Figure 2A, and Appendix A). In the fermented liquid of sour bamboo shoots, 1-octen-3-ol, 2,4-ditert-butylphenol, and D-limonene were the most dominant volatile compounds on the first day of fermentation. On the 7th day of fermentation, the content of volatile compounds sharply decreased, and some volatile compounds even disappeared, such as 1-octen-3-ol, linalool, 2-ethylhexanol, cyclooctanol, etc. As the fermentation time prolongs, the types and content of volatile substances gradually increase. On the 28th day of fermentation, 35 volatile compounds were detected, including 11 alcohols, 3 phenols, 6 ketones, 5 aldehydes, 6 esters, and 4 other compounds. Among these, 1-octen-3-ol, 2,4-ditert-butylphenol, and dimethylsilanediol had the highest content.

Throughout the process of fermentation, it was observed that three distinct compounds (1-octen-3-ol, 2,4-di-tert-butylphenol, and D-limonene) and their concentration exhibited the most prominent factor (Figure 2B). 1-Octen-3-ol (also known as mushroom alcohol) has a mushroom/earthy aroma [24], 2,4-di-tert-butylphenol has an almond/spicy flavor [25], and D-limonene is a terpene with a citrus-like flavor [26]. 2,4-di-tert-butylphenol is a product obtained from *Lactococcus* sp. with antifungal and antioxidant properties and has demonstrated its potential as a food additive that can improve food safety and promote health [27]. There is no clear understanding of the precursor or pathway that leads to this compound, even though the *Lactobacillus* gene clusters and enzymes for the shikimate and mevalonate pathways have been identified [28]. The latest research shows that 2,4-di-tert-butylphenol is a candidate compound for anti-diabetes-related enzymes [29]. In this study, 2,4-di-tert-butylphenol first decreased and then increased during the fermentation process, which may be related to the dominance of *Lactococcus* in the later stage. It is worth noting that p-cresol is considered a characteristic component in sour bamboo shoots [20], but this substance was not detected in our study, which may be related to factors such as materials and fermentation conditions. The content of alcohol volatile compounds was second only to that of phenols. With the passage of fermentation time, new alcohol volatile compounds were found, including 1-heptanol, 1-decanol, 1-octanol, and dimethylsilanediol. These are produced by Strecker degradation of amino acids through the Ehrlich pathway and by linoleic acid degradation [30]. Ester volatile compounds can exhibit a unique fruit aroma [31], as their lower threshold has a significant impact on flavor. Esters may be generated by the esterification reaction between alcohols and organic acids in sour bamboo shoots [32]. Due to different metabolic types, there are significant differences in the types of volatile aroma compounds produced compared with other studies [4]. Ketone volatile matter has a high threshold value and low content, rendering it minimally impactful to the flavor of the shoots. Maillard reaction, alcohol oxidation, or other acids may generate it under the influence of microorganisms [33]. As fermentation progresses, the content of aldehydes volatile compounds diminishes, with 2,4-dimethyl benzaldehyde exhibiting the highest content of aldehydes, and with volatile acid compounds present in very low concentrations. The results of this study are similar to those of Xu et al. in their study of the fermentation of Jipo bamboo shoots [34].

As shown in Figure 2C, principal component analysis was conducted on the physicochemical properties and volatile compounds of sour bamboo shoots. The analysis showed that PC1 and PC2 accounted for 46.0% and 29.0% of the total variance of the data, with a cumulative value of 75.0%. The distance between the first day of fermentation and other fermentation days is relatively far, indicating significant changes in physicochemical properties and flavor throughout the entire fermentation time. The distance between the 21st and 28th days of fermentation is close, indicating that their flavors are relatively similar. The pH value has a significant impact on flavor on the first day of fermentation, with representative volatile compounds such as ketones and phenols. The typical volatile compounds of sour bamboo shoots on the 7th day of fermentation were aldehydes, esters and others, which were closely related to the content of reducing sugar. The representative volatile compounds on the 14th day of fermentation were acids and closely related to the total acid content. On the 21st and 28th days of fermentation, the representative volatile compounds were alcohols and closely related to nitrite content. Some gram-negative bacteria (such as Enterobacteria) can convert nitrate into nitrite, and the changes in nitrite may be related to the changes in bacteria in the fermentation environment [6].

### 3.3. Odor Activity Value of Sour Bamboo Shoots

When sour bamboo shoots ferment, we found at least thirteen volatile compounds that we considered to be characteristic volatile compounds, differing from the results obtained by Chen, et al. [4]. In addition, the volatile compounds of sour bamboo shoots included, for the first time, 2,4-dimethylbenzaldehyde and 2,4-di-tert-butylphenol (Table 2). These differences may be caused by factors such as raw material type, raw material growth environment, and fermentation conditions.

The highest OAV (129.73~668.84) of benzaldehyde, 2,4-dimethyl-, which presents an almond/spicy flavor, was considered to be the volatile compound that contributes most to the flavor formation, followed by 2,2,4-trimethyl-1,3-pentanediol diisobutyrate (59.55~197.30), 3,6-nonadien-1-ol (1.29~80.66) and dibutyl phthalate (32.22~96.32). In contrast, phenols have the highest content, but their contribution to flavor is limited because of the high threshold. For example, 2,4-ditert-butylphenol dominates the content, with a high odor threshold (200 μg/kg) and limited contribution to the flavor. Phenol volatile substances typically produce unpleasant and irritating odors [25]. The OAV value of 3,6-nonadien-1-ol, which exhibits a fresh cucumber flavor, is higher on the 7th to 28th days of fermentation, making a critical contribution to the later flavor of sour bamboo shoots. Octanal and nonanal typically exhibit a fresh citrus flavor [35], but due to their low OAV value, they contribute less to the flavor of sour bamboo shoots. Nonanal was also found to be the main component of the sour bamboo shoot flavor by Guo, et al. [36]. Acetic acid, 1-heptanol, and 1-Octen-3-ol were characteristic flavor components, but their relative contents are relatively low, and their contributions to the flavor formation of sour bamboo shoots are limited. Among these, 1-octene-3-ol can be used as an edible essence, which has the aroma of mushroom house, lavender, rose or hay, and is consistent with the main flavor substances reported in other reports [4]. D-limonene and isophorone had high OAV values on the first day of fermentation, which may have contributed significantly to the early flavor formation of sour bamboo shoots.

**Table 2 foods-12-03035-t002:** Volatile compounds with OAV >1 in sour bamboo shoots during the 28 days of fermentation.

Compounds	Threshold (μg/kg)			OVA		
		1 d	7 d	14 d	21 d	28 d
1-Heptanol [37]	5.5	0.00	0.00	1.46	0.00	0.00
1-Octen-3-ol [24]	20	5.06	0.00	0.00	0.00	5.88
3,6-Nonadien-1-ol [24]	1.3	1.29	80.66	58.92	53.51	67.30
Silanediol, dimethyl- [24]	21	0.08	0.00	15.12	17.81	20.10
Isophorone [38]	2	35.08	10.53	0.00	0.00	3.91
Octanal [4]	0.7	3.58	7.53	2.23	2.04	3.32
Nonanal [4]	1	9.62	16.30	8.76	19.36	17.26
Benzaldehyde, 2,4-dimethyl- [37]	0.2	129.73	668.84	198.69	346.41	167.06
2,2,4-Trimethyl-1,3-pentanediol diisobutyrate [39]	0.13	59.55	197.30	131.84	113.28	111.13
Dibutyl phthalate [40]	0.26	59.07	50.91	96.32	52.67	32.22
D-Limonene [41]	10	35.42	1.04	0.24	0.00	0.32
2,4-di-tert-butylphenol [42]	200	11.25	6.77	7.09	8.46	9.70
Acetic acid [37]	5.5	0.00	0.00	5.69	0.00	0.00

### 3.4. Microbial Composition in Sour Bamboo Shoot

Alpha diversity analysis (coverage rate >99%), including Chao1, ACE, Shannon, and Simpson indices, can reflect the diversity and richness of microbial communities. On the first day of fermentation, the richness and diversity of microorganisms were the highest, which may be due to miscellaneous bacteria accompanying bamboo shoots entering the fermentation broth. On the other hand, infiltration may cause some nutrients from bamboo shoots to dissolve into the fermentation broth [43], providing sufficient energy sources for microbial growth and reproduction, and allowing aerobic microorganisms to proliferate in large numbers. The nutrient composition and oxygen decrease as the fermentation process progresses [44], and the growth and reproduction of aerobic microorganisms are inhibited. The dominant bacteria in the fermented liquid of sour bamboo shoots gradually give way to lactic acid bacteria, which rapidly reproduce and multiply, resulting in a higher level of microbial diversity on the 14th day [6]. On the 28th day of fermentation, the richness of microorganisms increased, similarly to the research results of Li, et al. [7] wherein the richness of microbial communities sharply increased on the 28th day of fermentation. PCoA and UPGMA analysis (Appendix A) found that there were significant differences in microbial structure in the early stages (1d and 7d) of fermentation, while fluctuations were relatively small in the later stages. This result may indicate that microbial succession is intense in the early stage, while dominant bacteria, composed mostly of lactic acid bacteria, form in the later stage [6]. Lactic acid bacteria not only reduce the pH value of the fermentation environment, but also produce bacteriocins to further inhibit the growth of miscellaneous bacteria [45].

#### 3.4.1. The Changes in Microbial Community Structure in Sour Bamboo Shoots during Fermentation

We employed high-quality sequencing reads to examine the bacterial community’s structure and dynamic succession at the phylum and genus levels during the fermentation of sour bamboo shoots. At the phylum level of bacteria, a total of 42 phyla were identified, and Firmicutes (20.02%), Proteobacteria (35.18%), and Bacteroidota (17.43%) were the predominant bacteria, accounting for 72.63% of the total bacterial population in the early stage of fermentation (Figure 3A). Then, Firmicutes (20.02–70.05%) increased significantly and dominated the subsequent fermentation stage, attaining the highest relative abundance on day 21 of fermentation. Proteobacteria is also the dominant phyla in the fermentation process, and its abundance varies from 27.93% to 35.18%. The results obtained are in concurrence with the research conducted by Chen, et al. [4], which revealed Firmicutes and Proteobacteria as the predominant phyla. In addition, Firmicutes was also found to be the dominant phyla in numerous traditional fermented vegetables, such as Jiangxi yancai [46], Suancai [47,48], and Kaili red sour soup [49].

At the genus level, the top 10 species at abundance levels in all samples were Enterococcus, Lactococcus, Enterococcus, Serratia, unclassified Enterobacteriaceae, Acinetobacter, Leuconostoc, Chitinophaga, Massilia, Enterobacter, and unclassified Enterobacterales (Figure 3B). Miscellaneous bacteria (81.59%) predominated on day 1 and decreased afterward. According to the findings of Guan, Huang, Peng, Huang, Liu, Peng, Xie and Xiong [6], the initial stage of sour bamboo shoot fermentation is characterized by the presence of five dominant genera, namely Lactococcus, Leuconostoc, Weissella, Enterobacter, and Raoultella. The variation in their abundance may be attributed to the origin of the raw materials used and could serve as an indicator of microbial diversity within the fermentation process. The abundance of Enterococcus in the sample increased significantly from an initial relative abundance of 0.59% to a peak level of 43.62% on day 7. Similarly, Lactococcus also showed a significant increase from the initial relative abundance of 1.07% to a peak level of 36.94% on day 21. According to prior investigations, it has been demonstrated that Lactococcus exhibits a stable function in the process of food fermentation [50,51]. Furthermore, in the Guangxi sour bamboo shoots, dominant genera were identified as Lactobacillus, Serratia, Stenotrichomonas, and Lactococcus [52].

#### 3.4.2. Representative Bacteria in Sour Bamboo Shoots

Linear discriminant analysis effect size (LEfSe) is a statistical method utilized for the identification of biomarkers by analyzing the variations within bacterial communities and recognizing distinctive features based on their abundances and associated categories. LEfSe with a threshold LDA score of 4.0 (Figure 4A,B) was set to distinguish and classify particular microbial taxa present at each stage of the fermentation process [6]. On the first day of fermentation, there were observable biomarkers from four phyla, seven orders, eight families, six genera, and four species. The sample from the 28th day of fermentation exhibited the presence of one order biomarker (Bacillales), one family biomarker (Bacillacea), and one genus biomarker (*Bacillus*), whereas only one genus of biomarkers (*unclassified Lactococcus*) was identified in the 14 day fermentation samples. According to previous studies, *Lactobacillus* is the dominant genus in the mid-to-late stages of fermentation [6]. Such differences may be caused by raw materials, production processes, geographical distributions, or analysis methods, among other things. Further comparative studies from different varieties and regions can be conducted in the future.

### 3.5. Correlations between Microbial Compositions and Flavor

During the fermentation process, physicochemical properties and main volatile compounds are closely related to the number and type of microbial communities [16]. Pearson correlation analysis was used to study the correlation between the physicochemical properties, characteristic volatile compounds (OAV > 1), and the dominant phyla and genera of the microbial community of sour bamboo shoots (Figure 5).

At the phylum level, Firmicutes was the overwhelmingly advantaged phylum. Firmicutes showed a significant negative correlation with D-limonene (R = −0.98), isophorone (R = −0.93), 2,4-di-tert-butylphenol (R = −0.82), and pH (R = −0.86). It was significantly positively correlated with 3,6-nonylidene-1-ol (R = 0.92), and positively correlated with 2,4-dimethyl benzaldehyde (R = 0.55), silanediol, dimethyl- (R = 0.52), nonanal (R = 0.59), and 2,2,4-trimethyl-1,3-pentylenediol diisobutyrate (R = 0.75). It has been reported by Chen, Li, Cheng, Liu, Yi, Chen, Wang and Cao [20] that Firmicutes can show significant direct correlations with fumaric acid, and geranyl acetone. In addition, a previous study has shown that the thick cell skin of Firmicutes can produce spores to resist extreme environments while being inhibited after the removal of D-limonene [53]. Proteobacteria was significantly positively correlated with 1-Octen-3-ol (R = 0.90), 2,4-di-tert-butylphenol (R = 0.99), and negatively correlated with three characteristic volatile compounds, namely, 2,4-dimethylbenzaldehyde (R = −0.66), 2,2,4-trimethyl-1,3-pentylenediol diisobutyrate (R = −0.84), and dibutyl phthalate (R = −0.40), indicating that proteobacteria was not conducive to the formation of the characteristic flavor of sour bamboo shoots. The majority of the genera attached to raw materials were Proteobacteria, such as *Pseudomonas*, *Vibrio*, and *Acinetobacter*. As a result, these were highly correlated with early fermentation. Fermentation conditions such as falling pH and increasing TA restricted most Proteobacteria genera [22]. Furthermore, it has been observed that Proteobacteria and Cyanobacteria display a positive correlation with textural attributes such as hardness, chewiness, and fracturability, suggesting that the bacteria belonging to the aforementioned phyla may have a crucial role in the alteration of cell wall properties [23].

In general, the genera *Enterobacter* was considered to be a spoilage bacterium in fermented vegetables [54], and exhibited strong positive correlations with two characteristic compounds (3,6-nonylidene-1-ol and 2,2,4-trimethyl-1,3-pentylenediol di-isobutyrate) in our study, suggesting that they may play a major role in suansun’s volatile flavor production (Figure 5B). Considering that *Enterobacter* is a foodborne pathogen, its presence in fermented bamboo shoots is undesirable. However, the edible safety of the final products of sour bamboo shoots should not be a concern: sour bamboo shoots are often cooked at a high temperature, which effectively reduces microbial contamination risk. It has also been reported that *Enterobacter* is responsible for the characteristic flavors found in sauerkraut [6], cheeses [55], and yucha [56], among other fermented foods. A common characteristic of lactic acid bacteria is that they produce acids from carbohydrate catabolism [45], while *Lactococcus* was significantly correlated with TA (R = 0.92) and pH (R = −0.94) in this study as well as with seven characteristic flavor substances. Some studies have shown that the fermented vegetable aroma was also produced by lactic acid bacteria, which are responsible for the production of esterases, lipases, and alcohol acetyltransferases, all of which enhance the flavor-forming qualities of fermented vegetables [57]. At the same time, *Serratia* had significantly positive correlations with silanediol, dimethyl- (R = 0.92) and TA (R = 0.89), and significantly negative correlations with hardness (R = −0.91), chewiness (R = −0.93), and fracturability (R = −0.91). Therefore, *Serratia* may affect the texture, a conclusion which is consistent with previously reported results [3].

## 4. Conclusions

Physicochemical properties, flavor compounds, and microbial composition of sour bamboo shoots produced in southwest China were correlated during the fermentation process (28 days). There was a decreasing trend in pH and TA values of fermentation liquid from sour bamboo shoots, followed by a stabilizing trend. It has been estimated that at least 13 volatile compounds (OAV > 1) are characteristic of sour bamboo shoots. As a result of HTS, it was found that *Enterococcus*, *Lactococcus*, and *Serratia* were the most dominant genera in sour bamboo shoots. Moreover, *Enterococcus* had significantly negative correlations with D-limonene and 2,4-dimethylbenzaldehyde, and significantly positive correlations with 3,6-nonylidene-1-ol and 2,2,4-trimethyl-1,3-pentylenediol di-isobutyrate. There were significant positive correlations between *Lactococcus* and silanediol, dimethyl- and TA, and significant negative correlations between *Lactococcus* and isophorone. *Serratia* had significantly positive correlation with silanediol, dimethyl- and TA, significantly negative correlation with texture properties (hardness, chewiness, and fracturability). A significant correlation was found between 3,6-nonylidene-1-ol and Firmicutes at the phylum level. The findings of this study can be valuable in enhancing the production of starters, which in turn can promote the industrial production of high-quality and safe sour bamboo shoots. However, there is still a lack of key flavor information and fermentation time, and further research is needed to obtain more comprehensive information to comprehensively evaluate the fermentation of sour bamboo shoots.

## Figures and Tables

**Figure 1 foods-12-03035-f001:**
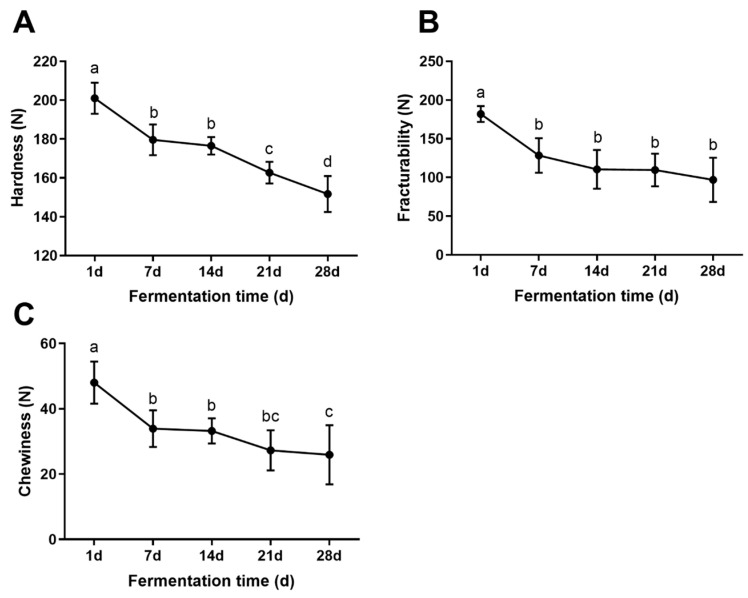
Texture properties of the sour bamboo shoot during the fermentation process. (**A**) hardness, (**B**) fracturability, (**C**) chewiness. Statistically significant differences are indicated by different lowercase letters at *p* < 0.05.

**Figure 2 foods-12-03035-f002:**
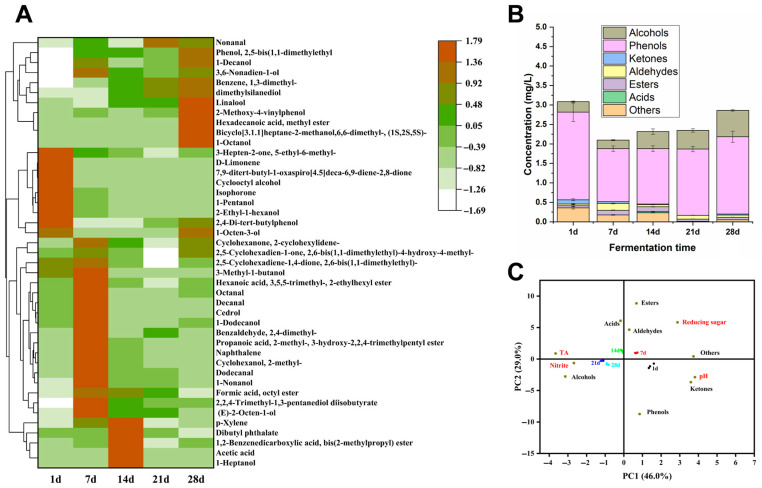
Relative contents of various volatile compounds (**A**), heatmap and dendrogram of the volatile compounds (**B**), and PCA analysis (**C**) in sour bamboo shoots during the 28 days of fermentation. Heatmap plots show the relative abundance of volatile compounds in samples (variables clustered vertically). Color intensity is proportional to volatile compound abundance.

**Figure 3 foods-12-03035-f003:**
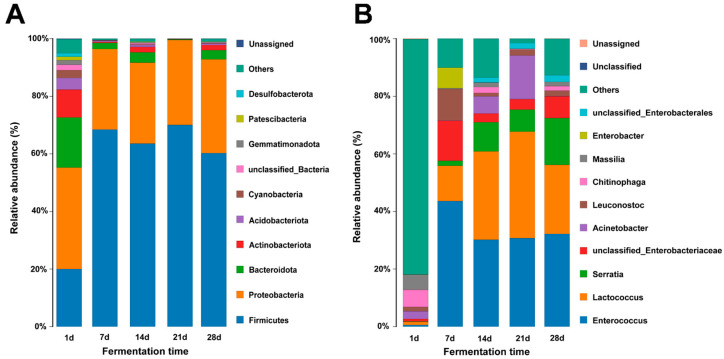
Changes of microbial community at the phylum (**A**) and genus (**B**) levels in sour bamboo shoots during the 28 days fermentation.

**Figure 4 foods-12-03035-f004:**
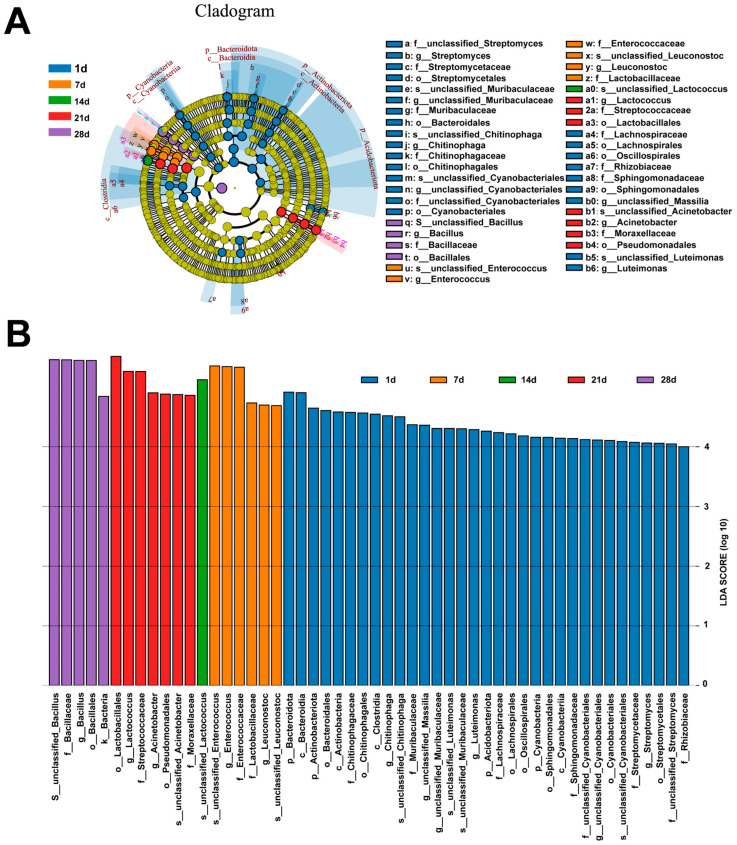
LEfSe of the microbial community during suansun fermentation at the OTU level. (**A**) lefse biomarkers cladogram, (**B**) lefse biomarkers.

**Figure 5 foods-12-03035-f005:**
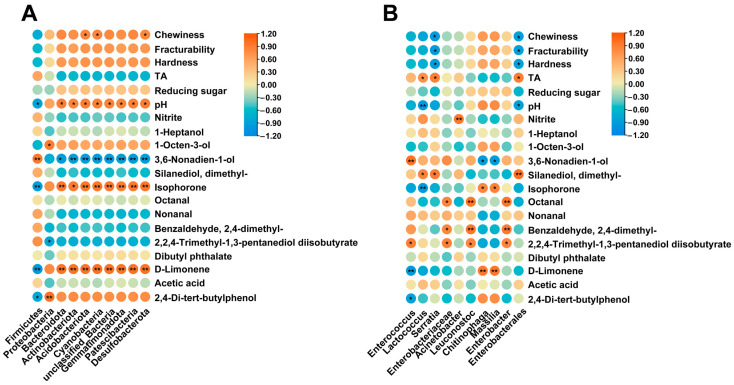
Pearson correlation analysis of the relative abundance of the top ten bacteria phylum levels (**A**) and genus levels (**B**) and variation of volatile compounds (OAV > 1) and physicochemical properties during sour bamboo shoot fermentation. * indicates significance at *p* < 0.05; ** indicates significance at *p* < 0.01.

**Table 1 foods-12-03035-t001:** Physicochemical properties of sour bamboo shoot liquids during the fermentation process.

	1 d	7 d	14 d	21 d	28 d
pH	5.73 ± 0.05 a	5.23 ± 0.14 b	4.92 ± 0.02 c	4.82 ± 0.04 c	4.81 ± 0.01 c
Titratable acidity (g lactic acid/L)	2.50 ± 0.43 e	3.75 ± 0.29 d	5.01 ± 0.29 c	7.5 ± 0.29 b	8.75 ± 0.29 a
Nitrite content (mg/kg)	0.12 ± 0.01 c	0.16 ± 0.01 bc	0.22 ± 0.04 b	0.74 ± 0.08 a	0.19 ± 0.03 bc
Reducing sugar (g glucose/L)	1.05 ± 0.05 b	1.08 ± 0.03 ab	1.18 ± 0.03 a	0.73 ± 0.06 c	0.63 ± 0.03 c

a–e Different lowercase letters in the same row indicate significant differences (*p* < 0.05).

## Data Availability

Data is contained within the article or Appendix A.

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
