# Peer review of "Dynamic Changes in Flavor and Microbiota in Traditionally Fermented Bamboo Shoots (Chimonobambusa szechuanensis (Rendle) Keng f.)"

_foods, 2023, doi:10.3390/foods12163035_

Round 1

Reviewer 1 Report

The manuscript is well-written in every aspect.

-Objectives are clearly stated and justified.

-The concept is novel, and it will certainly add to the scientific knowledge in the field.

-Organization of the manuscript is adequate. It is well written with satisfied language and grammar.

-The results are well argued and the conclusion drawn are realistic.

Following are some minor comments for improvement.

L88: Please provide more details of the sample.

L98: What was the fermentation condition?

L161: Please mention about PCA

L183: For how long is it safe?

L186: Supporting literature is missing.

L195-197: Please rewrite the sentence, as the meaning is not clear.

Fig 1: in the X axis unit is missing.

L239-245: Provide some literature support on the trend obtained.

L262: Pleas discuss more about the bi-plot and connect it with correlation matrix.

L273: Why 13 components are only considered to be characteristic volatile compounds during the fermentation process?

L306-308: Please rewrite.

L322: What might be the reason?

L366: What are the limitation of the study?

L387-388: Please mention the clusters and thereby assign the importances of the attributes

L402: Please support your trend with some literature.

L421-423: Are you concluding this from your own study?

L426: Please mention the limitations of the study.

Author Response

Response to Reviewers Comments

Manuscript No: foods-2521262

TITLE: Dynamic changes in flavor and microbial in traditionally fermented bamboo shoots (Chimonobambusa szechuanensis (Rendle) Keng f.)

Dear Reviewer:

Thank you very much for the consideration of our manuscript to be published in the Foods. I accept and appreciate your comments, which are all valuable and very helpful for revising and improving our paper, as well as the important guiding significance to our researches. We have studied comments carefully and have made a correction which we hope meet with approval. Revised portion is marked in red in the revised manuscript. The main corrections in the manuscript and the response to your comments are as flowing:

Point 1: Please provide more details of the sample (L88).

Response 1: As the reviewer’s suggestion, we have added the collecting time.

Point 2: What was the fermentation condition (L98)?

Response 2: Fermentation is carried out at room temperature, around 22±2℃.

Point 3: Please mention about PCA (L161).

Response 3: Thank you for your suggestion. We have mentioned about the PCA.

Point 4: For how long is it safe (L183)?

Response 4: The excessive marinating time is also likely to cause the microorganism content in the sour bamboo shoots to exceed the safety standards.  We will delve deeper into future research.

Point 5: Supporting literature is missing (L186).

Response 5:  Thank you for your careful guidance, we have corrected it.

Point 6: Please rewrite the sentence, as the meaning is not clear (L195-197).

Response 6: As per the reviewer’s suggestions, we have rewritten the sentence.

Point 7: in the X axis unit is missing (Fig 1).

Response 7: Many thanks for your careful guidance, we have added the X axis unit.

Point 8: Provide some literature support on the trend obtained (L239-245).

Response 8: As per the reviewer’s suggestions, we have provided some literature support on the trend obtained.

Point 9: Pleas discuss more about the bi-plot and connect it with correlation matrix (L262).

Response 9: As per the reviewer’s suggestions, we have added some discussion.

Point 10: Why 13 components are only considered to be characteristic volatile compounds during the fermentation process (L273)?

Response 10: We are very sorry that due to issues with our description, we have made modifications.

Point 11: Please rewrite (L306-308).

Response 11: As per the reviewer’s suggestions, we have rewritten it.

Point 12: What might be the reason (L322)?

Response 12We have added possible reasons as detailed in revised manuscript.

Point 13: What are the limitation of the study (L366)?

Response 13: We have added the limitation of the study as detailed in revised manuscript.

Point 14: Please mention the clusters and thereby assign the importance of the attributes (L387-388).

Response 14: As per the reviewer’s suggestions, we have mentioned the clusters.

Point 15: Please support your trend with some literature (L402).

Response 15: As per the reviewer’s suggestions, we have added the support literature.

Point 16: Are you concluding this from your own study (L421-423)?

Response 16We are very sorry that our conclusion is somewhat inappropriate, and we have deleted it.

Point 17: Please mention the limitations of the study (L426).

Response 17: As per the reviewer’s suggestions, we have added the limitations of the study.

We tried our best to improve the manuscript and made some changes in the manuscript. These changes will not influence the content and framework of the paper. And here we did not list the changes but marked in red in the revised manuscript. We appreciate for your warm work earnestly and hope that the correction will meet with approval.

Once again, thank you very much for your comments and suggestions.

Yours sincerely,

Qiyang Chen,

School of Life Science and Engineering,

Southwest University of Science and Technology,

E-mail: chenqiyang@swust.edu.cn

Reviewer 2 Report

Comment 1: line no 43-45 As you have mentioned “the sour bamboo shoots is to ferment by soaking them in mountain spring water or cold boiling water for certain periods” can you specify what is the exact period of fermentation in days/hours.

Comment 2: line no. 114 please mentioned the method of TA determination in brief

Comment 3: line no 117-118 “nitrate content in fermentation liquids was measured” please explain the method briefly as per given reference

Comment 4: line 120 “HS-SPME-GC-MS/MS” please expand at the first instance of mention, before using the abbreviation.

Comment 5: line no 142 to extract total genomic DNA from the fermented broth you have used CTAB method. Please clarify the broth can be used as it is for DNA extraction because during the fermentation the pH of the broth has reduced. If you have cultured prior to DNA extraction from the fermentation broth then mention the culturing conditions [medium and environmental conditions].

Comment 6: line no. 143 the obtained genomic DNA used for amplification but prior to amplification you must check the purity of the DNA, hence you must mention the purity of DNA.

Comment 7: line no. 147, 148 “PE250bp, 250bp” please provide the space between bp

Comment 8: line no. 186, 187 the sentence “the relatively lower peak reducing sugar content can be attributed to the breakdown of diverse saccharide degradation in the raw materials upon the initial day” need to more clarity please rewrite.

Comment 9: line no 259 “related to the content of reducing sugars” the “R” must be in lowercase

Comments 10: line no. 287 “odor threshold (200 ug/kg” please correct the unit replace u by µ. 

Overall English is fine.

Author Response

Response to Reviewers Comments

Manuscript No: foods-2521262

TITLE: Dynamic changes in flavor and microbial in traditionally fermented bamboo shoots (Chimonobambusa szechuanensis (Rendle) Keng f.)

Dear Reviewer:

Thank you very much for the consideration of our manuscript to be published in the Foods. I accept and appreciate your comments, which are all valuable and very helpful for revising and improving our paper, as well as the important guiding significance to our researches. We have studied comments carefully and have made a correction which we hope meet with approval. Revised portion is marked in red in the revised manuscript. The main corrections in the manuscript and the response to your comments are as flowing:

Point 1: line no 43-45 As you have mentioned “the sour bamboo shoots is to ferment by soaking them in mountain spring water or cold boiling water for certain periods” can you specify what is the exact period of fermentation in days/hours.

Response 1: In general, the fresh vegetable materials are immersed in mountain spring water or cold boiled water without any other spices, and then placed at ambient temperature to allow fermentation for 15–30 days in pickle jars (Guan et al., Food Research International, 2020).

Point 2: line no. 114 please mentioned the method of TA determination in brief

Response 2: As per the reviewer’s valuable suggestions, we have added the method of TA.

Point 3: line no 117-118 “nitrate content in fermentation liquids was measured” please explain the method briefly as per given reference

Response 3: Many thanks for your careful guidance, we have added the method of nitrate content briefly.  

Point 4: line 120 “HS-SPME-GC-MS/MS” please expand at the first instance of mention, before using the abbreviation.

Response 4: As per the reviewer’s valuable suggestions, we have provided its full name. 

Point 5: line no 142 to extract total genomic DNA from the fermented broth you have used CTAB method. Please clarify the broth can be used as it is for DNA extraction because during the fermentation the pH of the broth has reduced. If you have cultured prior to DNA extraction from the fermentation broth then mention the culturing conditions [medium and environmental conditions].

Response 5: We directly use bamboo shoot fermentation liquid to extract DNA. According to the TGuide S96 magnetic bead method (Chen et al., Food Bioscience, 2022), 200 µL of fermentation liquid was add to a 2 ml centrifuge tube, and then add 500 µL buffer SA, 100 µL buffer SC, and 0.25 g grinding beads.

Point 6: line no. 143 the obtained genomic DNA used for amplification but prior to amplification you must check the purity of the DNA, hence you must mention the purity of DNA.

Response 6: Many thanks for your careful guidance, we have added the content regarding the purity of the DNA.

Point 7: line no. 147, 148 “PE250bp, 250bp” please provide the space between bp

Response 2: As the reviewer’s valuable suggestions, we have added the space between bp.      

Point 8: line no. 186, 187 the sentence “the relatively lower peak reducing sugar content can be attributed to the breakdown of diverse saccharide degradation in the raw materials upon the initial day” need to more clarity please rewrite.

Response 8: Many thanks for your careful guidance, we have deleted this sentence.       

Point 9: line no 259 “related to the content of reducing sugars” the “R” must be in lowercase

Response 9: As the reviewer’s valuable suggestions, we have corrected it.         

Point 10: line no. 287 “odor threshold (200 ug/kg” please correct the unit replace u by µ. Response 10: Many thanks for your careful guidance, we have corrected the unit replace u by µ.         

We tried our best to improve the manuscript and made some changes in the manuscript. These changes will not influence the content and framework of the paper. And here we did not list the changes but marked in red in the revised manuscript. We appreciate for your warm work earnestly and hope that the correction will meet with approval.

Once again, thank you very much for your comments and suggestions.

Yours sincerely,

Qiyang Chen,

School of Life Science and Engineering,

Southwest University of Science and Technology,

E-mail: chenqiyang@swust.edu.cn

Reviewer 3 Report

Overall, the manuscript is interesting and brings new information on aromas and fermentation processes. The results need some further discussion to improve the manuscript, and some corrections need to be made.

1. Methodology. Please inform the volume of water used in the fermentation.

2. Methodology. How much volume was taken at each sampling time?

3. Results. Line 174. The authors stated that “The acidic environment during fermentation may be attributed to the presence of Bacillus and Lactobacillus”. Acidity is caused by acid or other acidic compounds and not due to the presence of microorganisms. The authors should correct this statement attributing the acidic environment to an acid present in the fermented broth. The microorganism may have produced this acid, but they are not acids.

4. Results. Line 178. Why did nitrite concentration increase? Please explain.

5. Figure 1. Please show the full error bars and not half error bars.

6. Results. Line 224. The authors stated that three distinct phenols were prominent, including 1-octen-3-ol and D-limonene; however, only 2,4-di-tert-butylphenol is a phenol. 1-octen-3-ol is an alcohol, and limonene is a terpene. Please correct.

7. Results. Line 227. D-limonene is not an aromatic alcohol and does not has a rose-like flavor. D-limonene is a terpene with a citrus-like flavor. Please correct and adjust your results accordingly.

8. Results. Line 270. This paragraph belongs to the methodology and not the results. Please inform the references used for the values of the odor threshold.

9. Results. The authors have described the changes in the volatile components, but little effort was made to discuss what caused these changes. What kind of chemical modifications were verified? Which kind of metabolic routes from the microorganisms consumed and produced the volatile compounds? Such discussion should have been made and would improve considerably the manuscript.

10. Results. In section 3.4, the authors have done a simple correlation analysis with the concentration of volatiles and microorganisms. However, how can they guarantee that these correlations are real? For example, did Firmicutes consume Limonene, or did another microorganism that was in its exponential growth phase consume Limonene? The complexity of fermentation, what each microorganism consumes to grow, what they excrete as defense mechanisms should be considered and makes these simple correlations to fail considerably. The authors should better explore the basic metabolism of each microorganism and revise this section.

Author Response

Response to Reviewers Comments

Manuscript No: foods-2521262

TITLE: Dynamic changes in flavor and microbial in traditionally fermented bamboo shoots (Chimonobambusa szechuanensis (Rendle) Keng f.)

Dear Reviewer:

Thank you very much for the consideration of our manuscript to be published in the Foods. I accept and appreciate your comments, which are all valuable and very helpful for revising and improving our paper, as well as the important guiding significance to our researches. We have studied comments carefully and have made a correction which we hope meet with approval. Revised portion is marked in red in the revised manuscript. The main corrections in the manuscript and the response to your comments are as flowing:

Point 1: Methodology. Please inform the volume of water used in the fermentation.

Response 1: Thank your careful guidance, we have added the volume of water (450 mL) (Line 97).

Point 2: Methodology. How much volume was taken at each sampling time?

Response 2: At each sampling time, three unopened bottles were randomly selected (Lines 100-101).

Point 3: Results. Line 174. The authors stated that “The acidic environment during fermentation may be attributed to the presence of Bacillus and Lactobacillus”. Acidity is caused by acid or other acidic compounds and not due to the presence of microorganisms. The authors should correct this statement attributing the acidic environment to an acid present in the fermented broth. The microorganism may have produced this acid, but they are not acids.

Response 3: Thanks for your comments and suggestions. We have changed the sentence as “The acidic environment during fermentation may be attributed to the acid or other acidic compounds produced by Bacillus and Lactobacillus” (Lines 175-176).

Point 4: Results. Line 178. Why did nitrite concentration increase? Please explain.

Response 4: As per the reviewer’s suggestions, we have explained the reason (Lines 183-189).

Point 5: Figure 1. Please show the full error bars and not half error bars.

Response 5: As per the reviewer’s suggestions, we have modified Figure 1 to show the full error bars in the revised manuscript.

Point 6: Results. Line 224. The authors stated that three distinct phenols were prominent, including 1-octen-3-ol and D-limonene; however, only 2,4-di-tert-butylphenol is a phenol. 1-octen-3-ol is an alcohol, and limonene is a terpene. Please correct.

Response 6: Thank you for your professional guidance. We have corrected our mistakes (Lines 236-237).

Point 7: Results. Line 227. D-limonene is not an aromatic alcohol and does not has a rose-like flavor. D-limonene is a terpene with a citrus-like flavor. Please correct and adjust your results accordingly.

Response 7: Thank you for your suggestion. We have made the necessary changes (Line 235).

Point 8: Results. Line 270. This paragraph belongs to the methodology and not the results. Please inform the references used for the values of the odor threshold.

Response 8: According to the reviewer's suggestion, we have moved this paragraph to Methods Section 2.7 (Lines 141-145) and added the references used for the values of the odor threshold.

Point 9: Results. The authors have described the changes in the volatile components, but little effort was made to discuss what caused these changes. What kind of chemical modifications were verified? Which kind of metabolic routes from the microorganisms consumed and produced the volatile compounds? Such discussion should have been made and would improve considerably the manuscript.

Response 9: Many thanks for your professional guidance, we have added corresponding discussions, as detailed in our revised manuscript.

Point 10: Results. In section 3.4, the authors have done a simple correlation analysis with the concentration of volatiles and microorganisms. However, how can they guarantee that these correlations are real? For example, did Firmicutes consume Limonene, or did another microorganism that was in its exponential growth phase consume Limonene? The complexity of fermentation, what each microorganism consumes to grow, what they excrete as defense mechanisms should be considered and makes these simple correlations to fail considerably. The authors should better explore the basic metabolism of each microorganism and revise this section.

Response 10: As the reviewer’s suggestions, we have rewritten this section.

We tried our best to improve the manuscript and made some changes in the manuscript. These changes will not influence the content and framework of the paper. And here we did not list the changes but marked in red in the revised manuscript. We appreciate for your warm work earnestly and hope that the correction will meet with approval.

Once again, thank you very much for your comments and suggestions.

Yours sincerely,

Qiyang Chen,

School of Life Science and Engineering,

Southwest University of Science and Technology,

E-mail: chenqiyang@swust.edu.cn